# Evaluation of a near-patient SARS-CoV-2 novel rapid diagnostic platform

Johannes C. Botha,[1,2] Karen Zafilaza,[3] Cathia Soulie,[3] Nicolas Yin,[4] Moira Spyer,[1] Sofia Balaska,[5] Stella Chatziioannidou,[6] Vaia Tsiakalou,[7] George Papadakis,[6] Lemonia Skoura,[5] Alexandros Zafiropoulos,[8] George Sourvinos,[8] Olivier Vandenberg,[9] Anne-Geneviève Marcelin,[3] Electra Gizeli,[7,10] Eleni Nastouli[1,2]

**ABSTRACT**   The goal of this study is to test a novel device and methodology based on the "Pebble" platform and real-time quantitative colorimetric loop-mediated isothermal amplification (qcLAMP) during SARS-CoV-2 detection using crude samples and extracted RNA. The new method employs an inexpensive lightweight device aimed toward rapid point-of-care testing. An extensive evaluation was performed consisting of 1,693 clinical samples across five independent clinical testing centers. Positive colorimetric results were observed within 20 minutes of testing. At a 20-minute time-to-positive cut-off, the specificity is 98.5% with a diagnostic accuracy of 91.9%, compared to qPCR assays. Our findings indicate that the SARS-CoV-2 qcLAMP diagnostic assay in conjunction with the Pebble device is ideal for point-of-care/near-patient testing.

**IMPORTANCE**   Here, we describe our analyses and validation of a novel real-time quantitative colorimetric loop-mediated isothermal amplification (qcLAMP) device, available under the name "Pebble" and associated SARS-CoV-2 diagnostic qcLAMP assay for clinical diagnostic use. The analyses were performed in five independent testing sites across Europe using clinical samples from the associated clinical sites and support the use of "pebble" and associated kit in the diagnostic environment.

**KEYWORDS**   SARS-CoV-2, colorimetric LAMP, near-patient, point-of-care, respiratory virus, diagnostics

R eal-time qualitative PCR (qPCR) has long been the cornerstone of molecular-based clinical diagnostics (1). The ability to detect and simultaneously further quantify nucleic acids highlights the significance of qPCR as a gold standard diagnostic tool (2). With this power of qPCR comes complexities such as the requirement of highly specialized and expensive equipment capable of maintaining precise temperature cycles with robust optical detection components and intricately designed fluorescent probes specific to the target nucleic acid (3, 4). Therefore, qPCR is largely used as a diagnostic tool in central high-throughput environments. The COVID-19 pandemic has highlighted the need and accelerated the development of alternative clinically significant diagnostic platforms (5).

The loop-mediated isothermal amplification (LAMP) assay is a less complex alternative to qPCR capable of detecting a broad range of targets without the need of expensive thermal cycling equipment (6–8). The LAMP assay is also faster and less sensitive to inhibition compared to qPCR (9, 10). Similar to qPCR, LAMP can be monitored in real-time through fluorescence or turbidimetry providing quantification data, but this requires bulky and expensive equipment (11, 12). More recent LAMP assays favor a naked-eye observable color change approach referred to as end-point colorimetric LAMP with simple means of maintaining the constant reaction temperature (13–17). The major

**Peer Reviewer** Arundhathi Venkatasubramaniam, Integrated Biotherapeutics Inc., Rockville, Maryland, USA

Address correspondence to Johannes C. Botha, j.botha@ucl.ac.uk, or Eleni Nastouli, e.nastouli@ucl.ac.uk.

The authors declare no conflict of interest.

drawback with naked-eye detection is that it requires highly experienced operators for result interpretation (13). A recently developed platform poses to overcome the mentioned shortcomings by offering real-time quantitative colorimetric LAMP (qcLAMP) in a near patient setting (18). The qcLAMP assay is performed in a small, inexpensive device offering stable isothermal reaction temperature control with real-time colorimetric detection synced directly to a smartphone. The low-cost and simple setup nature of this device is favorable for near patient and limited resource settings (18).

Here, we describe our analyses and validation of a novel real-time qcLAMP device, available under the name "Pebble" (BIOPIX DNA TECHNOLOGY, Gr), and associated SARS-CoV-2 diagnostic qcLAMP assay (CAT#000055 available at https://biopix-t.com/products/) for clinical diagnostic use. The analyses were performed in five independent testing sites across Europe using clinical samples from the associated clinical sites.

## MATERIALS AND METHODS

### Platform description

The platform comprises a small 18.1 cm (length) × 10.7 cm (width) × 5.2 cm (height) and lightweight (300 g) diagnostic system (Fig. 1A) that allows for the rapid molecular detection of infectious agents (such as COVID-19 and Influenza A) at or near point-of-care. The system performs a real-time qcLAMP assay by using a novel method for heating the testing tubes and detecting the results. Placing the six test tubes vertically and in close contact with a heating element allows efficient heating of the reaction sample for LAMP to take place while the walls of the Eppendorf remain free for side inspection (Fig. 1B). Following a simple sample-preparation step (Fig. 1C), few microliters of the diluted sample is added in an eppendorf tube with the LAMP reagents and placed inside the device in the special tube holder; operation of the device is controlled via a special smartphone app set-up (Fig. 1D). During each run, the color change of the reagent within the test tubes can, thus, be monitored via a digital camera placed opposite the test tubes. The camera collects images at predefined time intervals (10 s), analyses the red, green, and blue pixel values, and interprets the analysis into a graph (Fig. 1E). The graph shows the signal of the color-change (which is expressed as color index units) vs time (18). No internal calibration is required due to the nature of this device and the syncing of results to the online app. A first test-run is performed after setup to ensure the device operates as expected.

In the current study, the assay used in combination with the Pebble is a Nucleic Acid Amplification Test kit based on the qcLAMP method for the detection of SARS-CoV-2 N-gene RNA target. The CoV19 qcLAMP kit is compatible with but does not require RNA isolation and can be used directly with nasopharyngeal and oropharyngeal swab specimens in viral transfer media. Each kit comprises a 2× Enzyme mix consisting of a mixture of Bst polymerase and thermostable reverse transcriptase, optimized reaction buffer, Mg2+, and dNTPs. It further contains a 5× COV19 Primer mix which contains SARS-CoV-2 specific primers and hydroxynaphthol blue indicator, and a 5× Control Primer mix for amplifying a human endogenous target (RNase P). The supplied 2× BIOPIX sample buffer neutralizes common sample inhibitors for direct crude sample amplification and detection. The kit also includes mineral oil to prevent cross-contamination and minimize evaporation. The assay is performed on a Pebble qcLAMP Platform. A simple assay setup is required, outlined in the relevant kit SOP available at https://biopix-t.com/products/. Assay setup only requires micro-pipettes and small equipment available in most primary clinical laboratories.

For a test to be considered positive, an amplification curve should appear on the screen usually after 10 min from the set-up of the reaction. The change in the slope of the line at a specific time point indicates the presence of the target, as opposed to a line that remains almost flat throughout the monitoring period which indicates a negative sample.

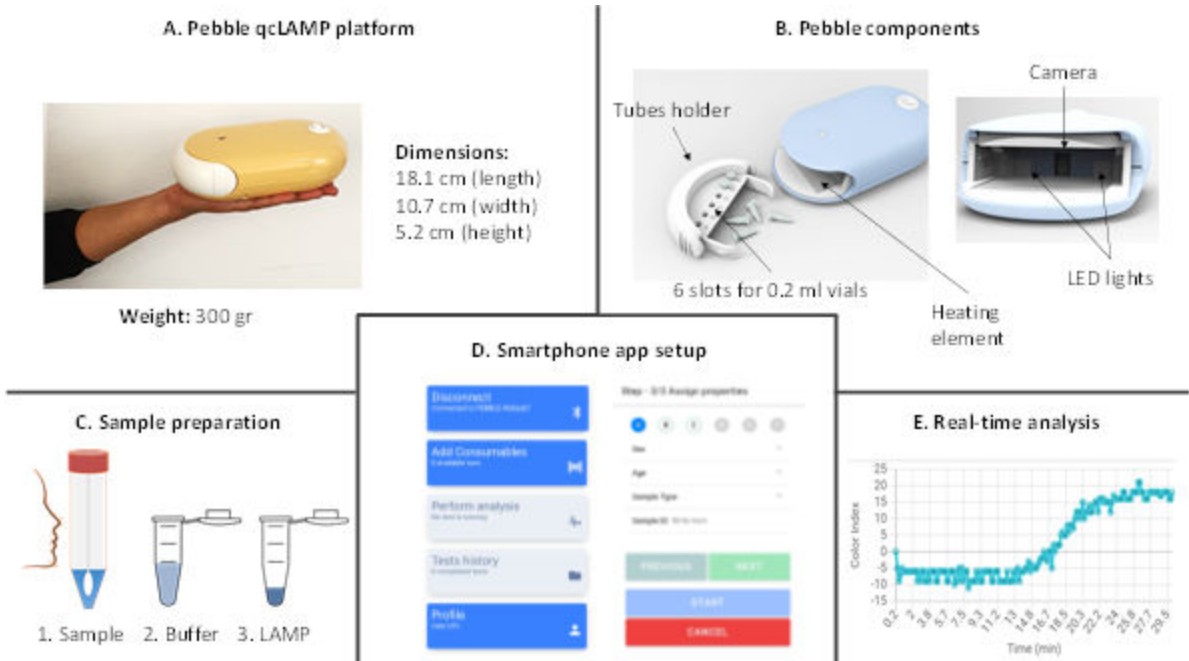

**FIG 1** Pebble point-of-care diagnostics for SARS-CoV-2. (A) The physical characteristics of the portable point-of-care platform. (B) Schematic of the Pebble device. The device consists of two parts; the tubes holder which has six slots for six independent LAMP tests and the main body which comprises all the electronics parts including a microprocessor that connects via Bluetooth to a smartphone and controls the operation of a heating element, a camera, and LED lights. (C) Schematic of the workflow. A swab (nasopharyngeal or saliva) is collected and dipped into 0.5–2 mL of a transfer medium, saline solution, or phosphate buffer saline (PBS). One hundred microliter of the medium is mixed with 100 µL of a neutralizing buffer provided in the kit box. Then 5 µL is added in the colorimetric LAMP reaction. (D) An Android application available in Google Play Store is used to control the device, monitor in real-time six tests simultaneously, and offer access to the data stored automatically in a cloud server. (E) A real-time curve that is displayed in the smartphone app during a test corresponding to a positive sample. The Y-axis depicts the change in the color of the reaction (Color Index Units) as a function of the test duration depicted in the X-axis (min).

## Study design and participants

The performance of the SARS-CoV-2 qcLAMP assay and associated Pebble device was assessed for diagnostic use with clinical samples in five independent clinical sites: University College London hospital (UCLH), UK; National Institute of Health and Medical Research Institut National de la Santé et de la Recherche Médicale UMR_S 1136 (INSERM), France; Brussels Academic Hospital Lab (LHUB-ULB), Belgium; University General Hospital of Heraklion (PAGNI), Greece; and AHEPA University Hospital of Thessaloniki, Greece by respective operators. Combined nose throat and saliva swab samples in viral transfer media were assessed in comparison with diagnostically used RT-PCR and the Fluidigm platform (19).

UCLH data at the peak of the COVID-19 pandemic in London, with a prevalence of 0.45 (0.41–0.50), suggested a clinical sensitivity of the screening assay ran by the standard of care RT-PCR of 0.86 (0.81–0.90). Sample size calculation (80% power, ß = 0.2, a = 0.05) for diagnostic accuracy of min accepted sensitivity for standard of care, i.e., 0.86 (expected 0.082) and specificity min 0.85 and expected 0.95: requiring 707 positive and 76 negative samples. This was calculated performed for the whole-evaluation study, i.e., we based it at UCLH as representative of what the INSERM and LHUB-ULB labs would also experience as very similar in terms of catchment area in a large metropolitan area. The above data were also representative for the other clinical sites during the pandemic.

### Nasopharyngeal swabs

For diagnostically significant analyses, we estimated a requirement to test at least 783 samples in total. In practice, a total of 1,637 samples, consisting of 279 RNA extracts, 523 fresh crude, and 714 frozen crude samples, were assessed among the five sites and used

to determine the sensitivity and clinical applicability of the qcLAMP assay performed on the Pebble device (Table 1). This included 121 SARS-CoV-2 negative, but positive for other respiratory viruses, samples used to assess cross reactivity (Table 2).

Evaluation swab samples were collected from symptomatic, asymptomatic, health-care workers, and self-referred individuals for UCLH, INSERM, LHUB-ULB, and PAGNI. A separate group of positive samples were patients or personnel who had symptoms and positive antigen test result. UCLH samples included both symptomatic patient samples as well as samples from the UCLH admission screening program. UCLH healthcare worker samples were not included. Nearly all positive samples in AHEPA University Hospital were healthcare workers tested during a screening program for the detection of SARS-CoV-2 by RT-PCR performed in saliva specimens.

SARS-CoV-2 qPCR negative samples were obtained from symptomatic and non-symptomatic individuals, and many samples were obtained as part of routine screening programs from non-symptomatic individuals. However, the possibility of infection with other respiratory pathogens should be assumed for a proportion of these samples (symptomatic individuals). All positive samples were stored at 4°C after qPCR, and qcLAMP was performed within 9 days. SARS-CoV-2 qPCR negative samples were also stored at 4°C after qPCR, and qcLAMP was performed within 15 days. Samples that were subsequently tested negative with qcLAMP were repeated with qPCR to verify results.

### Saliva samples

Saliva samples were tested only in AHEPA; collected specimens were dipped into 0.5 mL of PBS solution. Saliva samples were stable at room temperature for 120 h. Saliva samples were in parallel analyzed using the Fluidigm platform Advanta Dx SARS-CoV-2 RT-PCR Assay (19).

## RESULTS

### Analytical performance evaluation

Evaluation studies to assess the device sensitivity during *in vitro* analysis found the lower limit of detection to be 50 copies/µL, sensitivity, specificity, and precision of 95.8%, 96.7%, and 96.3%, respectively. These results were obtained by an independent accredited lab during the detection of SARS-CoV-2 RNA in compliance with EN 13612:2002.

### Clinical performance evaluation

#### Nasopharyngeal swabs

Clinical assessment was carried out in the four hospitals with swab samples collected in viral transfer media within a period of 2 weeks at the Omicron wave of the pandemic. Qualitative comparison analysis data of crude samples (diagnostic qPCR <31 Ct) were

TABLE 1   Clinical samples evaluated at four study sites

| Site | Type of sample | | | |
|------|------------------|------------------|-------------------|------------------|
|      | RNA extracts | Fresh crude[a] | Frozen crude[b] | Cross reactivity |
|      | ± | ± | ± | |
| PAGNI | 51/18 | 0/44 | 67/36 | |
| LHUB-ULB | 48/30 | 50/25 | 111/150 | |
| INSERM | 6/6 | 24/30 | 200/150 | 101 |
| UCLH | 120/0 | 200/150 | 0/0 | 20 |
| Total | 225/54 | 274/249 | 378/336 | 121 |
| Grand total | 279 | 523 | 714 | 121 |
| Overall | | | | 1,637 samples |

[a]Fresh crude: original sample aliquots stored at 4°C for less than 15 days.
[b]Frozen crude: original sample aliquots stored at −80°C more than 15 days.

**TABLE 2** Clinical samples for cross-reactivity analyses, SARS-CoV-2 negative but positive for other respiratory viruses

| Site | Positive viral target | | | | |
|------|-----------------------|--|--|--|--|
|      | Seasonal CoV | Influenza A | Respiratory syncytial virus | Adenovirus | Rhino/Enterovirus |
| INSERM | 62 | 20 (H3 + H1) | 20 | 0 | 0 |
| UCLH | 10 | 9 (H3) | 2 | 2 | 2 |
| Total | 72 | 29 | 22 | 2 | 2 |

used to determine the diagnostic sensitivity and specificity (with predictive values and concordance). Sensitivity and specificity were determined based on results after 20 and 30 minutes (min) of reaction time, respectively (Table 3). The combined sensitivity at 20 min was calculated as 80.3%, varying over the four testing sites from 73% to 87.5%. In comparison, the combined sensitivity at a 30 min cut-off was calculated as 85.4%, with testing site variation of 78%–92.9%. Combined specificity at a 20 min cut-off was determined to be 98.5%, 96.5%–100% variation over testing sites. A 30 min cut-off resulted in a reduction of the combined specificity to 94.2% where testing sites varied from 89% to 99.4%. This results in diagnostic accuracy of 91.9% (90.7%–93.2% over sites) at a 20 min cut-off and 91.5% (89.7%–93.5% over sites) at 30 min.

Cross-reactivity analysis performed at INSERM and UCLH included 121 clinical samples in total, 101 and 20, respectively. False positive SARS-CoV-2 signal was detected in 12/121 samples consisting of three Influenza, two Respiratory syncytial virus (RSV), one Enterovirus/RSV co-infection, and six seasonal CoV positives.

### Saliva samples

A total of 56 samples were analyzed, 35 positive samples and 21 negative samples. At the cut-off of 20 min, there were 10 false negative samples and 0 false positive identified. At the 30-min cut-off, there were six false negative and four false positive, respectively. Sensitivity ranged from 71.4% to 82.9%, specificity from 80.9% to 100%, and the accuracy was equal to 82.1%.

## DISCUSSION

An extensive clinical evaluation was performed in five independent clinical testing centers. In total, 1,693 clinical samples were analyzed in this diagnostic assay evaluation. This molecular diagnostic platform was assessed against standard-of-care molecular assays. This collaborative evaluation spread across five clinical sites (UCLH, INSERM, LHUB-ULB, PAGNI, and AHEPA) contributing 490, 517, 414, 216, and 56 samples, respectively. The Pebble device and associated SARS-CoV-2 assay are user-friendly

**TABLE 3** Comparison of sensitivity, specificity, and diagnostic accuracy between 20- and 30-minute test durations

| Site | Test duration (min) | Sensitivity (%) | Specificity (%) | Diagnostic accuracy (%) |
|------|---------------------|-----------------|-----------------|-------------------------|
| PAGNI | 20 | 82.4 | 100 | 90.8 |
|       | 30 | 85 | 99.4 | 92.9 |
| UCLH | 20 | 85.3 | 96.5 | 91.7 |
|      | 30 | 90.7 | 89 | 89.7 |
| INSERM | 20 | 73 | 99.5 | 90.7 |
|        | 30 | 78 | 98.5 | 91.7 |
| LHUB-ULB | 20 | 87.5 | 99.1 | 93.2 |
|          | 30 | 92.9 | 93.6 | 93.2 |
| Combined | 20 | 80.3 | 98.5 | 91.9 |
|          | 30 | 85.4 | 94.2 | 91.5 |

with easy-to-follow manufacturer-supplied guidance. This system is most suitable as a near-patient diagnostic solution and is not considered a high throughput platform. The time-to-positive values of the SARS-CoV-2 qcLAMP assay ranged from 8.2 min up to 20 min, with very few true positives detected after this. Most of the detected false positive results appear after 21 min of run time. Due to this observation, we considered two testing duration of 20 and 30 min. The 20-min cut-off (compared to 30 min) yielded an increase in specificity from 94.2% to 98.5% resulting in a diagnostic accuracy of 91.9%, whereas 30-min testing time increased the sensitivity to 85.4% from 80.3% (compared to 20 min). Furthermore, the minor variation observed in diagnostic accuracy between sites is minimized when using the 20-min testing time. Saliva sample testing resulted in lower assay sensitivity compared to nasopharyngeal swab samples, similar to other reports (20). Therefore, depending on use, we suggest taking the performance indicators into account for clinical interpretation.

In the point-of-care setting, the Pebble compares favorably with all relevant products that are already in the market, considering all aspects of the product, i.e., portability, cost, speed, and reliability. The first device for molecular diagnostics at the point-of-care with a complete proposition is the bench top device from Cepheid, "Genexpert I." The smallest version weighs over 8 kg, and it is being sold for $24,000, with each cartage (single test) being sold for over $40. The second alternative of "ID Now" from Abbott weighs 3 kg, the device costs $10,000, and each compatible test is sold between $50 and $100. Lastly, Roche offers Cobas Liat, which weighs 3.8 kg and costs $25,000 and $100 per test. The above systems, although simple, portable, rapid, and suitable for point-of-care, are designed and priced for healthcare providers operating in larger-scale facilities such as hospitals and clinics. They are still not attractive for doctors' offices or home testing. The other newer devices like the DNA nudge (21) or the Hibergene (22) have not been able to offer something significantly different in terms of size, weight, and cost, from the already established offerings. The Pebble for the point-of-care weighs under 0.3 kg, its price is €750 ($807), and each test is sold for €9 ($10). Its size, combined procurement, and running cost and its simplicity of operation will make it an attractive tool for near-patient diagnostics.

In the home testing environment, few platforms are available, as for example, the Acula/Silaris Dock and Lucira Health. Pebble can compete with the above technologies in terms of cost (no microfluidics, use of isothermal amplification instead of PCR, and no fluorescent dyes) and accuracy (quantitative vs qualitative results). In addition, when cleared for the home testing market, the method will compete with non-molecular tests, i.e., rapid tests (Antigen/Antigon) such as the BinaxNOW from Abbot or the Ellume health which might seem attractive to the end user due to lower cost, but these methods lack the reliability of molecular tests.

Pebble is intended to be used at point-of-care settings lacking the luxury of expensive and bulky pieces of equipment for performing pure nucleic acid amplification testing. These instruments (such as the ones used in the five hospitals of this study) although demonstrating high sensitivity and reliability, never reach these settings. The proposed method is offering an alternative solution for affordable and accessible molecular testing. Another advantage of Pebble is that it offers a test result within less than 30 min while typically the golden standard methods require hours to days for this. In settings where patients return for treatment, this is extremely important and could lead to the treatment of more patients than with a more sensitive test. This is a paradox identified in literature for more than 20 years ago which describes the counterintuitive discovery that a test with a sensitivity of 0.63 can be preferred to one with a sensitivity of 0.94 (23). Therefore, Pebble and qcLAMP are without any doubt an excellent choice in situations when patients may not return to the clinic for a second visit.

In conclusion, our study underlines the importance of comparing point-of-care/near-patient diagnostic tools to high-throughput laboratory-based instruments. Here, we provide evidence of the Pebble device in conjunction with the SARS-CoV-2 qcLAMP diagnostic assay kit for clinical diagnostic use. Our evaluation shows that this approach is

easy to use and robust and yields sensitive results, ideal for near-patient use in primary patient care facilities.

## ACKNOWLEDGMENTS

This work has received funding from the EC through the H2020-SC1-PHE-CORONAVI-RUS-2020–2B grant no. 101016083 (project acronym "IRIS-COV").

All authors contributed to the study conception and design. Material preparation and data collection and analysis were performed by J.C.B., K.Z., N.Y., S.B., S.C., V.T., L.S., and A.Z. The first draft of the manuscript was written by J.C.B. and reviewed by all authors. All authors read and approved the final manuscript.

## AUTHOR AFFILIATIONS

[1]Department of Infection, Immunity and Inflammation, Institute of Child Health, University College London, London, United Kingdom

[2]University College London Hospitals NHS Trust, Advanced Pathogen Diagnostics Unit, London, United Kingdom

[3]Sorbonne Université, INSERM, Institut Pierre Louis d'Epidémiologie et de Santé Publique, AP-HP, Hôpitaux Universitaires Pitié Salpêtrière-Charles Foix, Laboratoire de Virologie, Paris, France

[4]Department of Microbiology, Laboratoire Hospitalier Universitaire de Bruxelles–Universitair Laboratorium Brussel (LHUB-ULB), Université Libre de Bruxelles, Brussels, Belgium

[5]Department of Microbiology, AHEPA University Hospital, Medical School Aristotle University of Thessaloniki, Thessaloniki, Greece

[6]BIOPIX DNA TECHNOLOGY PC, Science and Technology Park of Crete, Heraklion, Greece

[7]Institute of Molecular Biology and Biotechnology, Foundation for Research and Technology-Hellas, Heraklion, Greece

[8]Laboratory of Clinical Virology, School of Medicine, University of Crete, Heraklion, Greece

[9]Research and Technology Innovation Unit, Laboratoire Hospitalier Universitaire de Bruxelles–Universitair Laboratorium Brussel (LHUB-ULB), Université Libre de Bruxelles, Brussels, Belgium

[10]Department of Biology, University of Crete, Heraklion, Greece

## AUTHOR ORCIDs

Johannes C. Botha ⓘ http://orcid.org/0000-0002-7635-7810
Nicolas Yin ⓘ http://orcid.org/0000-0003-1706-6869
Olivier Vandenberg ⓘ http://orcid.org/0000-0003-0484-4502
Eleni Nastouli ⓘ http://orcid.org/0000-0002-1684-2013

## AUTHOR CONTRIBUTIONS

Johannes C. Botha, Conceptualization, Data curation, Formal analysis, Investigation, Methodology, Validation, Visualization, Writing – original draft, Writing – review and editing | Karen Zafilaza, Data curation, Formal analysis, Investigation, Writing – review and editing | Cathia Soulie, Data curation, Formal analysis, Investigation, Writing – review and editing | Nicolas Yin, Investigation, Validation, Writing – review and editing | Moira Spyer, Project administration, Resources, Writing – review and editing | Sofia Balaska, Data curation, Investigation, Writing – review and editing | Stella Chatziioannidou, Formal analysis, Investigation, Methodology, Writing – review and editing | Vaia Tsiakalou, Data curation, Formal analysis, Investigation, Writing – review and editing | George Papadakis, Formal analysis, Methodology, Writing – review and editing | Lemonia Skoura, Data curation, Investigation, Writing – review and editing | Alexandros Zafiropoulos, Formal analysis, Resources, Writing – review and editing | George Sourvinos, Resources, Writing – review and editing | Olivier Vandenberg, Formal analysis, Resources, Writing – review and editing | Anne-Geneviève Marcelin, Conceptualization, Investigation, Resources, Writing – review and editing | Electra Gizeli, Conceptualization, Formal

analysis, funding acquisition, Project administration, Resources, Supervision, Writing – review and editing | Eleni Nastouli, Conceptualization, Formal analysis, Project administration, Resources, Supervision, Writing – review and editing

## DATA AVAILABILITY

Patient data are held at UCLH APDU, which encourages optimal use of data by employing a controlled access approach to data sharing, incorporating a transparent and robust system to review requests and provide secure data access consistent with the relevant ethics committee approvals. We will consider all requests for data sharing, which can be initiated by contacting Eleni Nastouli.

## ETHICS APPROVAL

All testing sites performed the evaluations following standard governance processes and permissions. At INSERM and LHUB-ULB, the study was approved by the local hospital Ethics committee. At UCLH, REC approval [R&D/Sponsor Reference Number(s):133228 IRAS 284088]. The current study has been approved by the Ethics Committee of PAGNI (decision number 560, 7 July 2021). The ehics committees of the other sites waived the need for having a separate ethical approval for using residual human body materials for the purpose of this study. In all cases, samples were fully anonymized.

## ADDITIONAL FILES

The following material is available online.

### Open Peer Review

**PEER REVIEW HISTORY (review-history.pdf).** An accounting of the reviewer comments and feedback.

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
