## [Reviewer comments · Microbiology Spectrum]

Microbiology Spectrum

Evaluation of a near-patient SARS-CoV-2 novel rapid diagnostic platform

Johannes Botha, Karen Zafilaza, Cathia Soulié, Nicolas Yin, Moira Spyer, Sofia Balaska, Stella Chatziioannidou, Vaia Tsiakalou, George Papadakis, Lemonia Skoura, Alexandros Zafiroopoulos, George Sourvinos, Olivier Vandenberg, Anne-Geneviève Marcelin, Electra Gizeli, and Eleni Nastouli

Corresponding Author(s): Johannes Botha, University College London

Review Timeline:

Submission Date:	March 15, 2024
Editorial Decision:	May 20, 2024
Revision Received:	July 23, 2024
Accepted:	August 20, 2024

Editor: Oliver Laeyendecker

Reviewer(s): Disclosure of reviewer identity is with reference to reviewer comments included in decision letter(s). The following individuals involved in review of your submission have agreed to reveal their identity: Arundhathi Venkatasubramaniam (Reviewer #1)

Transaction Report:

DOI: <https://doi.org/10.1128/spectrum.00672-24>

Re: Spectrum00672-24 (Evaluation of a near-patient SARS-CoV-2 novel rapid diagnostic platform)

Dear Dr. Johannes Christiaan Botha:

Thank you for the privilege of reviewing your work. Below you will find my comments, instructions from the Spectrum editorial office, and the reviewers comments.

There are some minor revisions needed.

Revision Guidelines

Sincerely,
Oliver Laeyendecker
Editor
Microbiology Spectrum

Reviewer #1 (Comments for the Author):

In lines 125-127, states UCLH data at the peak of the Covid-19 pandemic in London, with a prevalence of 0.45 (0.41,0.50), suggested a clinical sensitivity of the screening assay ran by the standard of care RT PCR of 0.86 (0.81, 0.90). It is not clear what the numbers in the brackets represent, please explain or include in a footnote if common terminology.

Reviewer #2 (Comments for the Author):

Botha et. al. provide a new methodology for the diagnosis of SARS CoV-2. Among the strengths of the study are the multiple sites at which the instrument was tested, the larger number of samples, and the advances in lowering the cost of the instrument while maintaining a high specificity and accuracy. This reviewer has little reservation with publishing the study as is.

Minor comments:

1. Could the authors discuss whether there were any particular procedural differences that caused variability between the different sites.
2. Could the authors comment on why in this assay the saliva samples seemed to work less well.
3. Would it be possible without sacrificing the authors' conclusions to standardize the currency that is used to compare the costs.

Evaluation of a near-patient SARS-CoV-2 novel rapid diagnostic platform

Review:

This paper reviews a novel device and methodology based on the Pebble platform and real time quantitative colorimetric LAMP for SARS-COV 2 detection in crude samples and extracted RNA. The loop-mediated isothermal amplification (LAMP) assay is a less complex alternative to qPCR capable of detecting a broad range of targets without the need of expensive thermal cycling equipment. The LAMP assay is also faster and less sensitive to inhibition compared to qPCR.

Pebble device in conjunction with the SARS-CoV-2 LAMP diagnostic assay kit was evaluated for clinical diagnostic use - Evaluation shows that this approach is easy to use, robust and yield sensitive results, ideal for near patient use in primary patient care facilities.

Decision – accept with revisions.

Comments and suggestions for the author

In lines 125-127, states UCLH data at the peak of the Covid-19 pandemic in London, with a prevalence of 0.45 (0.41,0.50), suggested a clinical sensitivity of the screening assay ran by the standard of care RT PCR of 0.86 (0.81, 0.90). It is not clear what the numbers in the brackets represent, please explain or include in a footnote if common terminology.

Confidential remarks for the editor

The manuscript is well written and easy to comprehend. The author's conclusions state that the Pebble platform in conjunction with LAMP assay is ideal for near patient use in primary patient care facilities, but do not address the false positives and false negatives issues seen with this assay. The authors need to add a couple of sentences addressing the false positives and negatives seen with this assay and any future work that may be done to address the specificity and sensitivity issues.

Reviewer #1 (Comments for the Author):

We thank the reviewer for such a positive review and comments. Below we have responded to the minor comment raised

In lines 125-127, states UCLH data at the peak of the Covid-19 pandemic in London, with a prevalence of 0.45 (0.41,0.50), suggested a clinical sensitivity of the screening assay ran by the standard of care RT PCR of 0.86 (0.81, 0.90). It is not clear what the numbers in the brackets represent, please explain or include in a footnote if common terminology.

We thank the reviewer for noting this omission. The numbers represent the median with the range in brackets and we have now replaced the “,” with “–” in order to clarify this.

Reviewer #2 (Comments for the Author):

Botha et. al. provide a new methodology for the diagnosis of SARS CoV-2. Among the strengths of the study are the multiple sites at which the instrument was tested, the larger number of samples, and the advances in lowering the cost of the instrument while maintaining a high specificity and accuracy. This reviewer has little reservation with publishing the study as is.

We thank the reviewer for this positive feedback and accepting our manuscript for publication without any major points noted. Below we have responded to all minor points raised.

Minor comments:

1. Could the authors discuss whether there were any particular procedural differences that caused variability between the different sites.

We thank the reviewer for raising this point and we believe with that the manuscript is now improved with our clarification on the minor inter-laboratory variation in accuracy. The assay itself and the setup required are “pretty straight forward” in accordance with the aim to develop an easy to use diagnostic, All sites received the same instructions and training' and scientists had virtual and face to face meetings. We believe, therefore, that the minor variation in diagnostic accuracy is mostly due to variability in sample conditions (collection, sample age, storage/transport conditions). This minor variation is further reduced when using a testing time of 20min.

We have now added the below in the manuscript (lines 214-215): **Furthermore, the minor variation observed in diagnostic accuracy (believed to be due to possible**

differences in storage time and conditions rather than factors associated with the assay performance) is minimised when using the 20 min testing time”.

2. Could the authors comment on why in this assay the saliva samples seemed to work less well.

We thank the reviewer for giving us the opportunity to add a reference for this point. Early in the pandemic, there was hope of saliva being an easier sample to collect. However, this was proven more difficult both at collection point as well as testing stage and saliva samples are reported to be of poorer overall diagnostic performance compared to nasopharyngeal swabs.

We have now made an addition in the manuscript (lines 215-217) and added a relevant reference: “Saliva sample testing resulted in lower assay sensitivity compared to nasopharyngeal swab samples, similar to other reports” (20).

20. Stokes W, Berenger BM, Portnoy D, Scott B, Szelewicki J, Singh T, Venner AA, Turnbull L, Pabbaraju K, Shokoples S, Wong AA, Gill K, Guttridge T, Proctor D, Hu J, Tipples G. 2021. Clinical performance of the Abbott Panbio with nasopharyngeal, throat, and saliva swabs among symptomatic individuals with COVID-19. *European Journal of Clinical Microbiology & Infectious Diseases* 40:1721–1726.

3. Would it be possible without sacrificing the authors' conclusions to standardize the currency that is used to compare the costs.

Thank you for this note, we have now included the \$ equivalent of the Pebble costs.

Re: Spectrum00672-24R1 (Evaluation of a near-patient SARS-CoV-2 novel rapid diagnostic platform)

Dear Dr. Johannes Christiaan Botha:

Your manuscript has been accepted, and I am forwarding it to the ASM production staff for publication. Your paper will first be checked to make sure all elements meet the technical requirements. ASM staff will contact you if anything needs to be revised before copyediting and production can begin. Otherwise, you will be notified when your proofs are ready to be viewed.

Sincerely,
Oliver Laeyendecker
Editor
Microbiology Spectrum